# Confident or Seek Stronger: Exploring Uncertainty-Based Small LM Routing From Benchmarking to Generalization

## Abstract

Small language models (SLMs) are increasingly deployed on edge devices for personalized applications, offering efficient decoding latency and reduced energy consumption. However, these SLMs often generate inaccurate responses when handling complex queries. One promising solution is uncertainty-based SLM routing, offloading high-stakes queries to stronger large language models (LLMs) when resulting in low-confidence responses on SLM. This follows the principle of *If you lack confidence, seek stronger support* to enhance reliability. Relying on more powerful LLMs is yet effective but increases invocation costs. Therefore, striking a routing balance between efficiency and efficacy remains a critical challenge. Additionally, efficiently generalizing the routing strategy to new datasets remains under-explored. In this paper, we conduct a comprehensive investigation into ***benchmarking and generalization of uncertainty-driven routing strategies from SLMs to LLMs over 5000+ settings***. Our findings highlight: *First*, uncertainty-correctness alignment in different uncertainty quantification (UQ) methods significantly impacts routing performance. *Second*, uncertainty distributions depend more on both the specific SLM and the chosen UQ method, rather than on downstream data. Building on the insight, we propose a proxy routing data construction pipeline and open-source a hold-out set to enhance the generalization on predicting the routing curve for new downstream data. Experimental results indicate that proxy routing data effectively bootstraps routing performance without any new data. The source code is available at https://anonymous.4open.science/r/quodlibeta.

## 1  Introduction

Large language models (LLMs) deployment on edge devices has gained increasing attention in recent years, primarily due to their potential for low-latency, privacy-preserving inference. Given the computational and memory constraints of edge devices, small language models (SLMs) (e.g., Phi2-mini [35] or Llama3.2-3B [70] are designed for resource-efficient deployment, particularly on devices such as smartphones and wearable devices. Their overarching goal is to democratize the deployment of LMs, making it accessible and affordable to users across diverse settings and at any time [52, 86, 83]. However, these SLMs often lack the robustness and scalability of LLMs [8] (e.g., GPT-4o [2] and Llama-3.1-405B), especially when faced with diverse and complex input queries under the deployment on edge devices, which eventually degrade the overall performance. This limitation raises a critical need for exploring solutions to increase the response reliability of SLMs.

To mitigate this unreliability, a line of work proposes to partially offload challenging and complex queries from SLMs to LLMs [11, 59, 32, 66]. A hybrid system is then established to wisely route the queries from SLMs and seek more reliable and deterministic responses from stronger LLMs.

Table 1: Uncertainty quantification (UQ) methods evaluated in our benchmark. "Model Access" specifies whether a method views the LM's weights/logits (white-box) or only its generated output (black-box). "Require Training?" indicates if additional training is needed. See Subsection 2.1 for taxonomy details and Subsection 3.1 for method descriptions.

| Uncertainty Quantification (UQ) Methods | Taxonomy | Model Access | Require Training? |
| --- | --- | --- | --- |
| Average Token Prob [53] | Token/sequence probabilities | White-box | No |
| $p$(True) [39] | Token/sequence probabilities | White-box | No |
| Perplexity [21] | Token/sequence probabilities | White-box | No |
| Jaccard Degree [47] | Output consistency | Black-box | No |
| Verbalization-1s [76, 69] | Verbalized uncertainty | Black-box | No |
| Verbalization-2s [69] | Verbalized uncertainty | Black-box | No |
| Trained Probe [4, 39, 53] | Uncertainty probe | White-box | Yes |
| OOD Probe [39, 53] | Uncertainty probe | White-box | Yes |

Although LLMs can exhibit superior performance, they incur high maintenance and inference costs given the large scale of model size and their infrastructure (i.e., a single NVIDIA A100 GPU can cost approximately $2,000 per month for deployment). Inaccurate routing by SLMs increases the volume of queries forwarded to LLMs, necessitating greater bandwidth allocation for maintaining the service of LLMs. As a result, operational costs and budgetary requirements rise accordingly, especially when continuous deployment is required. Hence, developing an effective routing strategy is crucial for fully deploying SLMs [59, 66, 11], as it both enhances response reliability and reduces the costs associated with services and data transmission.

Leveraging SLMs' self-uncertainty estimation emerges as a robust strategy for enhancing routing effectiveness [11, 16]. By relying on the self-assessed uncertainty, the system can better decide whether to handle a query locally or delegate it to a larger model without the aid of extra routers, ensuring that only queries deemed unreliable by the SLMs are routed to LLMs. As a result, the uncertainty-based routing approach not only generalizes well to new datasets, as only self-assessed information from SLM is needed, but it also reduces the high operational costs associated with accurately running LLMs. To this end, we aim to explore two open and nontrivial research questions for uncertainty-based SLM routing:

*1) What is the best practice of uncertainty estimation for query routing from SLMs to LLMs?*
In this research question, we benchmark the uncertainty-correctness alignment of each uncertainty quantification (UQ) method under its impact on SLM routing. A good alignment is a key factor for successful routing decisions, as any misalignment can cause unnecessary offloading with extra cost. However, SLMs may struggle to provide reliable uncertainty estimates [33, 15, 73], making them less effective as indicators for query routing. Thus, we benchmark the alignment between uncertainty and correctness, paving the insights for establishing more effective routing strategies[1].

*2) What is the best practice to initially establish an effective routing strategy when generalizing to new datasets?* In this research question, we explore how to generalize routing strategies to new datasets. Existing approaches [59, 32] rely on sufficient new downstream data to make routing decisions for optimal performance-cost trade-offs, but this process is time-consuming and labor-intensive. Broadly speaking, collecting and analyzing full downstream datasets under varying SLM configurations can be prohibitively costly, delaying implementation, which is not practical in real-world scenarios. This delay is particularly problematic in high-stakes scenarios, such as medical wearable devices, where reliability is critical, and inaccuracies are unacceptable even in early deployment stages. Based on our findings, we provide a data construction pipeline to predict the routing curves in new downstream scenarios without any new downstream data. A generated proxy routing dataset as a data-agnostic hold-out set enables the estimation of effective routing decisions via the predicted routing curves. We further benchmark the benefits of this proxy routing dataset, demonstrating its generalization ability in predicting the routing curve to new datasets.

This work offers an accessible and reproducible pipeline for uncertainty-based routing from benchmarking to generalization. Our main contributions are summarized as follows:

---

[1]For the convenience of writing, we interchangeably use uncertainty and confidence, where low uncertainty refers to high confidence.

- **Comprehensive benchmarking and detailed analysis:** This benchmark evaluates 8 UQ methods across 14 datasets to examine the alignment between uncertainty and correctness in routing tasks. We incorporate 8 SLMs and 2 LLMs to emulate real-world deployment scenarios. We then delve into key observations from the extensive results and conclude the insights for developing uncertainty-based SLM routing.

- **Proxy routing data for generalizing routing to new data:** Building on our benchmarking pipeline, we introduce a proxy routing data construction pipeline designed to generalize the routing curve prediction in new downstream scenarios. Empirical results show that this proxy routing data generalizes effectively the routing prediction to new datasets *without relying on any new downstream data*.

## 2 Reviewing Different Schools of Uncertainty Quantification and LLM Routing

### 2.1 Uncertainty Quantification for LMs

Uncertainty quantification methods estimate a model's confidence in its predictions [31]. For traditional classification and regression, uncertainty estimation is well-established [23]. However, for LLMs generating free-form responses to complex queries, estimating uncertainty is more challenging because the output space can grow exponentially with vocabulary size, and each sequence spans multiple tokens [20]. Existing uncertainty quantification approaches for LLMs can be grouped into the following four categories.

**Via verbalizing uncertainty.** This line of work prompts language models to report linguistic confidence [53, 56]. To enable LMs to verbalize confidence, researchers have proposed fine-tuning them to express uncertainty [46] or teaching them to verbalize confidence through in-context learning [17]. Verbalized confidence can take the form of linguistic expressions of uncertainty or numerical scores [24]. Multiple studies find that LLMs tend to be overconfident when reporting confidence [76, 69]. To mitigate this overconfidence, prompting strategies such as multi-step elicitation, top-$k$, and Chain-of-Thought [72] have been explored [69]. Sampling multiple response-confidence pairs and designing more effective aggregation strategies can also help mitigate overconfidence [76]. Moreover, [69] reports that verbalized confidence is typically better calibrated than the model's conditional probabilities.

**Via analyzing token/sequence probabilities.** This line of research derives confidence scores from model logits for output tokens [24, 33, 38]. The confidence of a generated sequence is computed by aggregating the log-probabilities of its tokens. Common aggregation strategies include arithmetic average, minimum, perplexity, and average entropy [20, 21, 71]. Because not all tokens in a sequence equally reflect semantic content, SAR reweights token likelihoods to emphasize more meaningful tokens [18]. However, different surface realizations of the same claim can yield different probabilities, implying that the calculated confidence reflects how a claim is articulated rather than the claim itself [53]. To combine LM self-assessment with token probabilities, $p$(True) is proposed: the model is asked whether its generated response is correct, and the probabilities of True/False tokens serve as the confidence score [39, 69].

**Via gauging output consistency.** This line of research (e.g., SelfCheckGPT [54]) assumes that high-confidence LLMs produce consistent outputs [53]. A typical approach samples $m$ responses for a given input query, measures inter-response similarity, and calculates a confidence score from meaning diversity [20]. Common ways to measure pairwise similarity include Natural Language Inference (NLI) and Jaccard similarity [24]. Consistency is then assessed by analyzing the similarity matrix, for instance, by counting semantic sets, summing eigenvalues of the graph Laplacian or computing eccentricity [47]. Because different sentences can express the same meaning, semantic entropy [40] first clusters responses by semantic equivalence before measuring consistency.

**Via training uncertainty probes.** This approach trains classifiers to predict whether an LLM will arrive at the correct answer for a particular query, using predicted probabilities as confidence scores [24]. Training data is often obtained by sampling multiple answers per question at a fixed temperature and labeling each for correctness [39]. A probe (commonly a multi-layer perceptron) then takes hidden states as inputs to predict correctness [4, 42]. Because in-domain training data is not always available, Contrast-Consistent Search trains probes unsupervisedly by maximizing

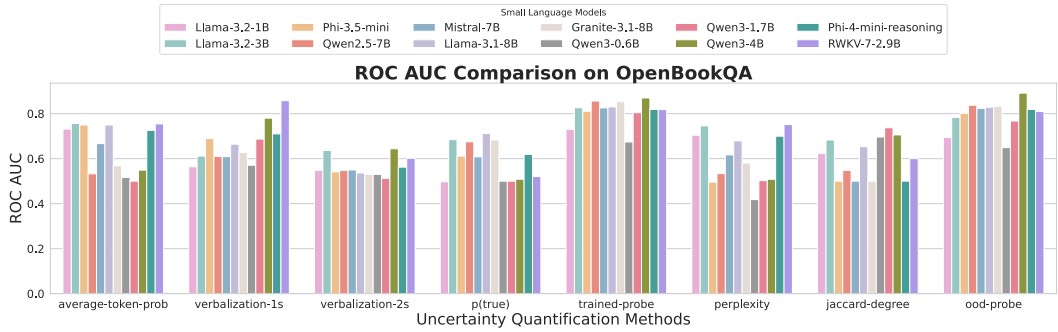

Figure 1: The ROC AUC scores measure the alignment between confidence and correctness across different SLMs and uncertainty quantification methods on OpenBookQA. A higher ROC AUC indicates a stronger alignment.

representation distances between contradictory answers on Yes/No questions [7]. Furthermore, whether probes trained on out-of-distribution data remain effective is still under debate [39, 53, 40].

## 2.2 LLM Routing

In query-routing scenarios, recent approaches train additional classifiers to direct queries to different SLMs or LLMs based on historical performance metrics and user feedback data [16, 59, 66, 37, 84]. For instance, RouterBench [32] collects inference outputs from selected LLMs to aid in the development of routing classifiers. However, these methods face significant challenges when encountering new downstream tasks, as such data falls outside the distribution of the existing training data. This limitation makes them less practical for real-world scenarios, such as on personal edge device deployment, where adaptability to unseen conditions is crucial. Our work focuses on how to establish routing systems between SLMs and LLMs and generalize to new downstream tasks. In this manner, uncertainty-based routing is an appropriate solution to overcome these challenges, as uncertainty is directly extracted from SLMs themselves. Furthermore, we propose a proxy routing data construction pipeline to initialize a routing system that generalizes to unseen datasets.

## 3 Benchmarking Uncertainty-based SLM Routing

In this section, we systematically evaluate 12 SLMs and 4 LLMs on 15 datasets using 8 UQ methods (see Table 1) for uncertainty-based SLM routing. This section details the datasets, models, and UQ methods, followed by several key findings and practical considerations. All experiments are conducted on four 80GB NVIDIA A100 GPUs.

### 3.1 Benchmark Coverage and Setup

**Language Models.** We evaluate 12 open-source SLMs, organized into three categories: non-reasoning LMs, reasoning LMs, and a recurrent neural network (RNN) model. The non-reasoning models are Llama-3.2-1B-Instruct [55], Llama-3.2-3B-Instruct [55], Phi-3.5-mini-instruct [1], Mistral-7B-Instruct-v0.3 [36], Qwen2.5-7B-Instruct [78], Llama-3.1-8B-Instruct [19], and Granite-3.1-8B-Instruct [26]. The reasoning models are Qwen3-0.6B, Qwen3-1.7B, Qwen3-4B, and Phi-4-mini-reasoning [77]. The RNN model is RWKV-7-2.9B [61]. These SLMs come from Alibaba (four models), Meta (three), Microsoft, Mistral AI, IBM, and LF AI & Data. Except for RWKV-7-2.9B, all adopt decoder-only Transformer architectures and are available on Hugging Face. We also include four LLMs: three open-source models—Llama-3.1-70B-Instruct [19], Qwen3-32B, and DeepSeek-R1 [29]—and one proprietary API model, GPT-4.1 mini [34]. Qwen3-32B and DeepSeek-R1 are reasoning LLMs, whereas Llama-3.1-70B and GPT-4.1 mini are non-reasoning.

**Datasets.** Experiments span 15 datasets from four domains: *(1) Mathematical Reasoning* (AQuA [48], GSM8K [13], MultiArith [63], SVAMP [60], MATH-500 [43]), *(2) Commonsense Reasoning* (CommonsenseQA [67], HellaSwag [80], OpenBookQA [57], PIQA [6], TruthfulQA [45], Wino-Grande [64], BoolQ [12], Social IQa [65]), *(3) Conversational and Contextual Understanding*

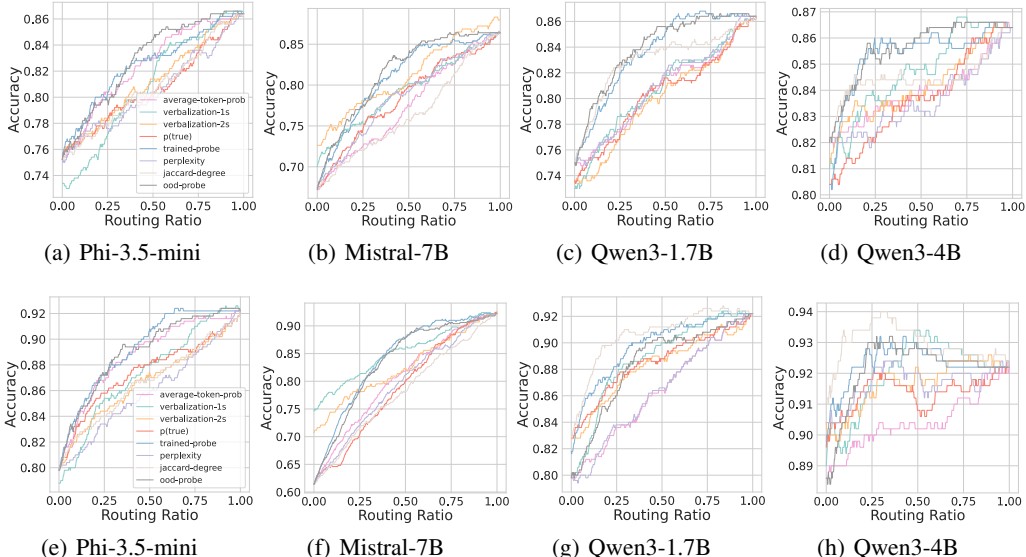

Figure 2: Overall accuracy vs. routing ratio with different UQ methods and SLMs. (a)-(d) show the results of routing to DeepSeek-R1 on the CommonsenseQA dataset; and (e)-(h) demonstrate the results of routing to GPT-4.1 mini on the OpenBookQA dataset.

(CoQA [62]), and *(4) Problem Solving* (MMLU [30]). These cover free-form, multiple-choice, and True/False question answering and are available via Hugging Face. Table 2 in Appendix A provides further details.

**UQ Methods and Hyperparameters.** We evaluate 8 approaches from the four categories in Section 2.1. *(1) Average token probability* uses the probability of the chosen option token (e.g., "A") for multiple-choice tasks or the mean probability of all generated tokens for free-form tasks. *(2) Perplexity* is computed for a sequence of $N$ output tokens $\{y_i\}_{i=1}^{N}$ with probabilities $\{p(y_i)\}_{i=1}^{N}$ as $\exp\left(\frac{1}{N}\sum_{i=1}^{N}\ln p(y_i)\right)$, and its reciprocal serves as the confidence score. *(3) p(True)* is a method where the LM first outputs an answer, then evaluates the generated response using only "True" or "False." The probabilities for these two tokens are normalized to sum to 1, and the probability of "True" is used as confidence. *(4) Verbalized confidence in a single response* (denoted as verbalization-1s) prompts the model to output both the answer and numeric confidence in one step. *(5) Verbalized confidence in the second round* (denoted as verbalization-2s) obtains the confidence in a separate, follow-up query after the model has provided an answer. *(6) The degree matrix* (denoted as jaccard-degree) generates $m = 5$ samples (temperature $1.0$) for one query, computes pairwise Jaccard similarities, and sets confidence to $\mathrm{trace}(mI - D)/m^2$, where $D$ is the degree matrix. *(7) Trained probe* is a four-layer MLP with LeakyReLU activations, trained on a fixed subsample of the in-domain training set for each dataset, taking as input the hidden states from the eighth-to-last transformer layer. We train for $20$ epochs (learning rate $5 \times 10^{-4}$). *(8) Trained probe on out-of-distribution data* (denoted as ood-probe) is identical in architecture but trained on all other datasets. e.g., if AQuA is evaluated, the ood-probe is trained on the remaining 14 datasets (20 epochs, learning rate $1 \times 10^{-4}$).

For verbalization-based methods, we discard queries when the model does not follow instructions to produce a confidence score. For free-form question answering, we use GPT-4.1 mini to evaluate whether a response is essentially equivalent to the ground truth answer [85].

## 3.2 Report Observations

In this section, we present our benchmarking results analyzing the impact of uncertainty-correctness alignment on routing tasks. More observations and experimental results on proxy routing and routing can be found in Appendix C.1.

**Observation ❶: Uncertainty estimation in SLMs may exhibit misalignment with prediction correctness.** From the theoretical perspective, well-calibrated uncertainty scores do not necessarily

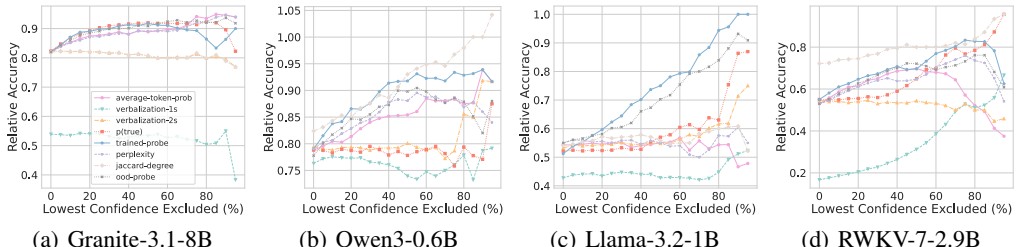

(a) Granite-3.1-8B     (b) Qwen3-0.6B     (c) Llama-3.2-1B     (d) RWKV-7-2.9B

Figure 3: Relative accuracy of SLMs vs. LLMs on top-$k$% confident queries. "Relative accuracy" is the ratio of SLM accuracy to LLM accuracy. The x-axis "Lowest Conf. Excluded" shows the percentage of low-confidence queries removed; for example, 80 means 80% of queries with the lowest confidence are excluded, leaving the top 20%. (a) and (b) compare SLMs to Llama-3.1-70B on GSM8K, while (c) and (d) compare SLMs to Qwen3-32B on BoolQ.

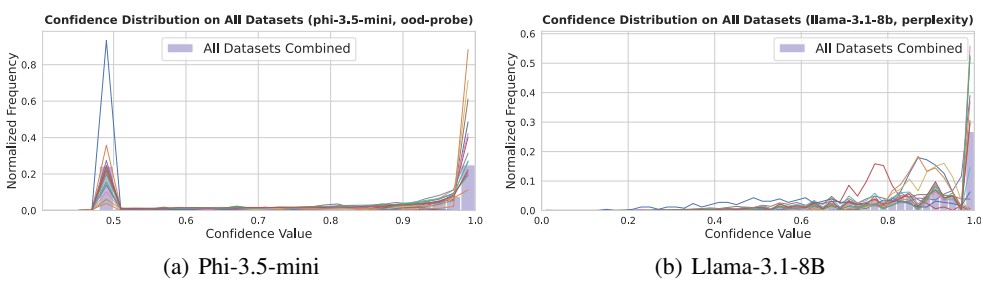

(a) Phi-3.5-mini         (b) Llama-3.1-8B

Figure 4: Confidence distributions across 15 datasets. The histogram depicts the aggregated distribution from all datasets, while each curve represents a single dataset. (a) Confidence of Phi-3.5-mini by OOD Probe; (b) Confidence of Llama-3.1-8B by Perplexity.

imply a strong correlation with the correctness of the predictions [33, 11]. The predictions of models might be perfectly calibrated yet still display relatively low accuracy (i.e., confidently provide wrong answers). This phenomenon is also evident in our benchmark results (illustrated in Figure 1). We compute AUC scores to quantify the correlation between extracted uncertainty and prediction correctness, treating correctness as a binary ground truth and using confidence values as the ranking metric. The results show that not all UQ methods effectively exhibit a strong alignment between confidence and prediction correctness. Moreover, from Figure 1 and Figure 8, we can observe that the alignment may vary across datasets for the same SLM and UQ method. For instance, Perplexity [21] demonstrates strong alignment for Phi-3.5-mini on the MultiArith dataset but fails on the OpenBookQA dataset. On the other hand, OOD Probe, Trained Probe, and Perplexity obtain consistently decent alignment compared to other UQ methods across different SLMs and domains of datasets. Conversely, we notice that verbalization-based methods, namely verbalization-1s [69, 53], and verbalization-2s [69], consistently withhold low alignment between uncertainty and prediction correctness. More experimental results can be found in Appendix C.1.

**Observation ❷: Verbalization-based UQ methods struggle to extract uncertainty in SLMs for query routing.** We find that verbalization methods like verbalization-2s [69] obtain poor alignment between confidence and prediction correctness, and this misalignment can lead to inferior routing performance in SLMs, where the conclusion can be found in Figure 2. Recent advancements [75, 79] also show that uncertainty scores derived from verbalization may exhibit good reflection on models' intrinsic uncertainty of prediction across multiple models and datasets. This discrepancy poses a significant challenge for establishing effective routing performance since queries that are actually correct may be unnecessarily routed from SLMs to LLMs, thereby increasing the overall cost of deploying routing systems.

**Observation ❸: A good routing standard highly depends on UQ methods with good uncertainty-correctness alignment.** A notable phenomenon occurs when UQ methods, such as Trained Probe [53], exhibit strong alignment, leading to significant improvements in routing performance. This is because

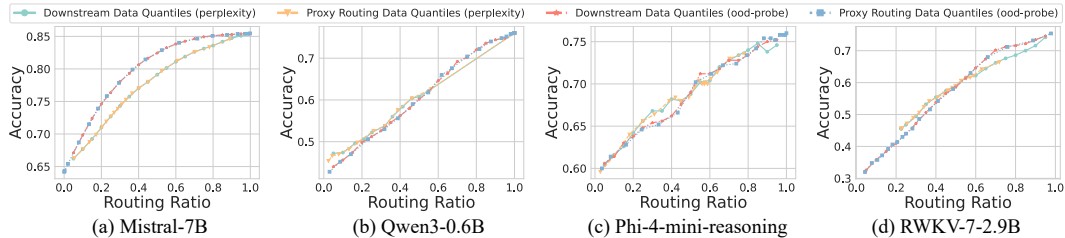

Figure 5: Routing results from four SLMs to Llama-3.1-70B on HellaSwag, with the remaining 14 other datasets constituting the proxy routing data.

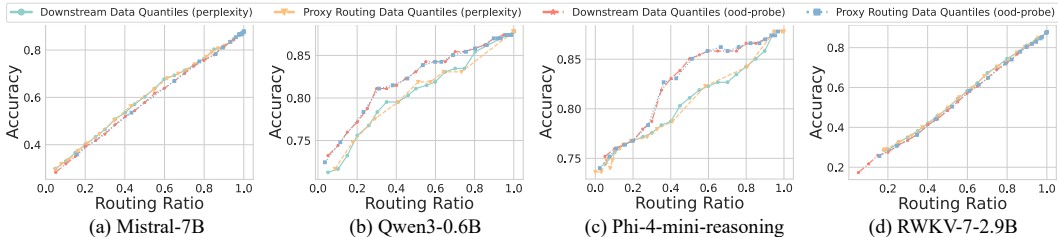

Figure 6: Routing results from four SLMs to DeepSeek-R1 on AQuA (mathematical reasoning), with eight commonsense-reasoning datasets and one conversational & contextual understanding dataset constituting the proxy routing data, demonstrate the strong generalization capability of the proxy routing data in predicting the routing curve.

the extracted uncertainty scores from these UQ methods more effectively indicate whether SLMs produce correct predictions. Among all UQ methods evaluated for routing tasks, we find that Trained Probe [53], OOD Probe [39, 53], and Perplexity [20] consistently rank as the top three methods for SLM routing. Therefore, a comprehensive analysis of UQ methods before deploying a routing system in SLMs is highly recommended to ensure efficient query routing.

**Observation ❹: SLMs can match LLM performance on high-confidence queries.** Although SLMs generally underperform LLMs, we find that for queries where SLMs exhibit high confidence, their accuracy approaches that of LLMs. To illustrate, we progressively remove queries starting from those with the lowest SLM confidence and compute the ratio of SLM to LLM accuracy on the remaining top-$k\%$ queries (Figure 3). As more low-confidence queries are excluded, SLMs achieve comparable performance to LLMs. For instance, on GSM8K, Qwen3-0.6B achieves performance nearly equal to Llama-3.1-70B on the top 20% highest-confidence queries. Moreover, the effectiveness of this selection depends on the uncertainty quantification (UQ) method: approaches with stronger alignment (e.g., Trained Probe [53]) yield higher relative accuracy than weaker ones (e.g., verbalization-2s) across all query exclusion rates. Additional results appear in Appendix C.2.

# 4 Generalizable SLM Routing for New Downstream Scenarios

In this section, we first describe the pipeline for constructing proxy routing data with experimental details. We then investigate how well the proxy routing data can predict the routing curve for new downstream scenarios without accessing the new datasets. Finally, we discuss our results and offer several insights into the proxy routing data for establishing routing in early-stage deployments.

## 4.1 Proxy Routing Data Construction Pipeline

We aim to evaluate the effectiveness of proxy routing data in generalizing the routing curve predictions to new downstream scenarios, without relying on additional downstream data. Specifically, the proxy routing data serves as a data-agnostic hold-out set tailored to a particular SLM, which can generalize its routing standards across various new downstream datasets. By leveraging this proxy routing data, we establish a generalizable routing framework for the routing deployments in the new scenario.

---
**Algorithm 1** Proxy Routing Data Construction Pipeline
---
**input** A collection of datasets $\mathbb{D} = \{\mathcal{D}_i\}_{i=1}^N$ with $N$ domains
**output** A set of proxy routing data $\boldsymbol{X}$
  1: Collect diverse domain of dataset $\mathcal{D}_i$ to form $\mathbb{D} = \{\mathcal{D}_i\}_{i=1}^N$
  2: Generate uncertainty distributions $\{\mathcal{F}_\mathbb{D}\}_{i=1}^M$ of $\mathbb{D}$ with selected UQ methods
  3: Sample $\boldsymbol{X} = \{x_j \mid x_j \in \boldsymbol{X}_i \sim \{\mathcal{F}_\mathbb{D}\}_{i=1}^M \ \forall i, j\}$ from $i$-th bin in $\{\mathcal{F}_\mathbb{D}\}_{i=1}^M$
---

This approach simplifies deployment of routing systems by eliminating the need for dataset-specific routing analysis and demonstrates that proxy routing data can generalize across diverse datasets.

The overall construction pipeline is detailed as follows. Let $\mathbb{D} = \{\mathcal{D}_i\}_{i=1}^N$ be a diverse collection of datasets, where $N$ denotes the number of distinct domain types included in the collection. We select a diverse collection of datasets $\mathbb{D}$ with various domains, such as commonsense reasoning, mathematics, and more, where we follow the settings in [50]. And then, we process every data instance in $\mathbb{D}$ through selected UQ methods to capture their corresponding uncertainty distributions $\{\mathcal{F}_\mathbb{D}\}_{i=1}^M$ with $M$ bins, where $M \in \mathbb{Z}^+$ is an arbitrary number. These distributions serve as the sampling foundation of each data instance in forming proxy routing data. Finally, data instances obtained in the set of proxy routing data $\boldsymbol{X}$ are weighted-sampled from each bin of $\{\mathcal{F}_\mathbb{D}\}_{i=1}^M$ such that $\boldsymbol{X} = \{x_j \mid x_j \in \boldsymbol{X}_i \sim \{\mathcal{F}_\mathbb{D}\}_{i=1}^M \ \forall i, j\}$. This ensures similar distribution in proxy routing data across various uncertainty levels presented in $\{\mathcal{F}_\mathbb{D}\}_{i=1}^M$. The resulting collection of these sampled data instances forms the final proxy routing dataset. The detailed pipeline of constructing the proxy routing dataset is outlined in Algorithm 1.

## 4.2  Proxy Routing Data Setups

**Benchmark Settings.** We evaluate the constructed proxy routing data on 15 SLMs and 4 LLMs across 15 datasets. Based on the observations and results from the previous benchmark section, we select 2 UQ methods that demonstrate the strongest alignment between predicted uncertainty and actual correctness: "OOD Probe" [39, 53] and "Perplexity" [20] method. We consider the routing performance evaluated on the entire new dataset as the ground truth. To simulate new dataset scenarios, we introduce two evaluation settings: (1) fully out-of-domain and (2) partially in-domain. First, for the out-of-domain setting, we evaluate a target dataset using proxy routing data derived from source datasets with no domain overlap. Second, in the partially in-domain setting, we designate one dataset as the target and construct its proxy routing data using the remaining 14 datasets, where the domain of the dataset may partially overlap. The target dataset's generalization performance is then evaluated using this proxy routing set, which does not contain any information from the target dataset. All reported results represent the average across three individual experimental runs.

**Data Construction Settings.** The proxy routing data is weighted-sampled from each bin of the proxy routing data distributions, with the number of bins set to 30. We sample $10\%$ of the instances from each bin to form the final proxy routing data. The temperature is fixed at 0 with a fixed random seed of 50 to ensure reproducibility.

## 4.3  Routing Curve Prediction with Proxy Routing Data

We provide several key insights into the generalization ability of proxy routing data as follows.

**Insights ❶: The extracted confidence distribution is predominantly determined by the chosen SLM and uncertainty quantification (UQ) method, with minimal dependence on the downstream dataset.** As illustrated in Figure 4, confidence scores aggregated from 15 different tasks exhibit a nearly identical shape regardless of the specific dataset. Instead, they vary notably with different SLMs and UQ methods. This finding suggests that the confidence distribution is largely data-agnostic, enabling the construction of proxy routing data that generalizes to new tasks without any new datasets.

**Insights ❷: Proxy routing data helps SLM routing to predict an accurate routing curve without any new data, allowing routing strategies to be initialized on SLMs without accessing new datasets.** Building on our findings about uncertainty distributions, we sampled a data subset to create a final proxy routing dataset using the pipeline described in Section 4.1. We then utilized this proxy routing dataset to predict all thresholds for different routing ratios in new downstream scenarios.

The experimental results (see Figure 5 and Figure 6) show that the routing curves from the proxy routing data closely match those from the entire new downstream dataset in both evaluation settings, indicating that the proxy routing data provides strong capability for establishing routing strategies on unseen downstream datasets. An identical phenomenon is observed across multiple UQ methods and different SLMs, highlighting the potential of proxy routing data to initiate the routing process for any new dataset, independent of the UQ method or SLM used. More results are in Appendix C.3.

## 5 Challenges and Opportunities

❶ **How to cash-in routing efficiency on new edge devices?** Based on the benchmark results, proxy routing data provides a robust foundation for establishing routing policies on new edge devices without accessing prior knowledge at the early stage of deployment. This enables the routing policies with strong generalization to new dataset scenarios and enhances the efficiency across diverse deployments for personal edge devices. While proxy routing data holds a good performance in the early deployment stage, an important direction to explore is how to effectively leverage additional private on-device data to strengthen the quality of proxy routing data, aiming to continuously enhance the deployment of personalized routing strategies. With the aid of proxy routing data, less private data is required, but striking a balance between privacy and performance remains an open challenge.

❷ **How to effectively strike a balance between LLM routing efficiency and utility?** We empirically observe that by leveraging UQ methods with strong uncertainty-utility alignment (e.g., Perplexity and OOD Probe methods), routing thresholds can effectively be determined with the sweet points of efficiency and utility. However, achieving such sweet spots can be challenging due to the variability in downstream datasets and the sensitivity of UQ methods to LLM-specific characteristics. Additionally, discrepancies across different device types, such as variations between iOS and Android systems , further complicating the process, requiring tailored strategies and analytics to account for platform-specific constraints and capabilities. Based on these factors, providing a fair apple-to-apple comparison regarding routing performance is inherently challenging. Researchers should be mindful of these complexities and focus on developing methods that are not only efficient but also capable of handling long-context scenarios effectively.

❸ **How is the performance when conducting compression (e.g., pruning, quantization) on the on-device model?** As with the on-device models discussed in the above sections, we directly adopt a pre-trained small model without any modifications. Alternatively, on-device models can also be generated by compressing larger models. Specifically, numerous works have explored methods for compressing LLMs into smaller sizes using techniques such as pruning [22] and quantization [74, 44]. The advantage of employing compression methods is that the smaller models compressed from larger ones tend to retain similar distributions of the output, thereby mitigating the issue of distribution shift.

❹ **Uncertainty-aware routing in on-device multimodal language models.** While LLMs typically operate with a single modality for both input and output, a promising research direction involves exploring uncertainty-aware routing in multimodal language models (MLLMs). For instance, in vision-language models (VLMs) such as LLaVa [49] and InternVL [9], the inputs include both images/videos and text. By incorporating visual modalities, the properties of vision tokens significantly influence the output. As a result, the uncertainty in the generated text differs from that of language-only models. Benchmarking and generalizing uncertainty-aware routing for on-device MLLMs is a valuable direction for the research community.

## 6 Conclusion

This paper investigates the routing accuracy of SLMs in estimating their uncertainty and establishing best practices for initiating effective routing strategies. Through comprehensive benchmarking of 15 SLMs, 4 LLMs, 8 UQ methods, and 15 datasets across 5000+ settings, we found that the alignment between uncertainty and correctness significantly impacts routing performance. Additionally, our experiments show that uncertainty distributions depend primarily on the specific SLM and UQ method rather than the downstream data. Building on the insights, we introduced a proxy routing data construction pipeline and a hold-out dataset to generalize routing strategies without prior knowledge of new downstream data. The results confirm that the proxy routing data effectively bootstraps routing, indicating its strong potential for benefiting in resource-efficient SLM deployment.

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
