# OpenReview forum: "Confident or Seek Stronger: Exploring Uncertainty-Based Small LM Routing From Benchmarking to Generalization"
_NeurIPS.cc/2025/Conference — Submitted to NeurIPS 2025_

### Official Review · Reviewer_2XVr · 2025-06-30

**Clarity:** 3
**Significance:** 3
**Originality:** 2
**Rating:** 4
**Confidence:** 2

**Summary:**

This paper investigates the use of uncertainty-based routing from small language models (SLMs) to large language models (LLMs) as a practical approach for improving reliability in edge-device deployment scenarios. It presents a comprehensive benchmark over 5000+ settings involving 15 datasets, 12 SLMs, 4 LLMs, and 8 uncertainty quantification (UQ) methods. The authors identify critical alignment patterns between uncertainty and correctness that affect routing efficacy. To tackle the challenge of generalizing routing strategies to unseen tasks, the authors propose a proxy routing data construction pipeline, enabling routing threshold estimation without accessing new downstream data. The proposed framework addresses an important efficiency–reliability trade-off in real-world LLM deployment.

**Questions:**

Have the authors evaluated existing multi-LLM routing systems under real or simulated on-device conditions, specifically in terms of latency and memory usage?

**Ethical Concerns:**

["NO or VERY MINOR ethics concerns only"]

**Final Justification:**

The authors address most of my concerns (comparison with baselines & performance on code-related tasks) during the rebuttal phase. Hence, I maintain a borderline accept recommendation.

**Limitations:**

Yes.

**Paper Formatting Concerns:**

NA.

**Quality:**

3

**Strengths And Weaknesses:**

***Strengths:***
1. The paper is well-organized and clearly written, especially the motivation and experimental sections.

2. The experimental evaluation is thorough. This paper conducts a comprehensive empirical evaluation across a wide range of models, datasets, and UQ methods.

***Weaknesses:***
1. It is unclear how existing baseline methods—such as RouterBench—perform in unseen domain settings. A direct comparison in Figures 5 and 6 would greatly help contextualize the benefit of the proposed proxy routing strategy.

2. It remains unclear how well the approach generalizes to code-related tasks, which are a common and important application domain for LLMs. More experiments on HumanEval or MBPP datasets are expected to be conducted.

---

> ### Author Rebuttal · Authors · 2025-07-31
>
> Thank you for your tremendous efforts in reviewing our paper. We would like to further answer your questions as follows.
>
> **[Q-1] Results on Code-Related Tasks**
>
> **[A-1]** We ran additional experiments on **MBPP** (Python coding problems). Below we report ROC AUC (uncertainty–correctness alignment) for two SLMs across 8 UQ methods. Trained Probe and OOD Probe perform best for Mistral‑7B, while Jaccard Degree and OOD Probe perform best for Qwen3‑0.6B.
>
> *Table 1: Uncertainty–Correctness Alignment on MBPP*
> ||AverageTokenProb|P(True)|Perplexity|Jaccard Degree|Verbalization-1s|Verbalization-2s|Trained Probe|OOD Probe|
> |--|--|--|--|--|--|--|--|--|
> |Mistral-7B|0.6060|0.5305|0.6110|0.5|0.5352|0.5|**0.6527**|**0.6220**|
> |Qwen3-0.6B|0.5657|0.5|0.5800|**0.6911**|0.5039|0.5778|0.4655|**0.6338**|
>
> Next, we evaluate our **Proxy Routing Data** on MBPP. Since we are not able to provide figures like Figure 5 and Figure 6 here, we compute the average root mean squared (RMS) distance between the routing curve derived from the full downstream dataset and the curve obtained using our method. Lower RMS distances indicate better performance. Below we show results when routing from Mistral‑7B and Qwen3‑0.6B to GPT‑4.1 mini, using two UQ methods (OOD Probe and Perplexity). We observe that **our method predicts accurate routing thresholds on MBPP without accessing it at all**, leading to small RMS distances.
>
> *Table 2: Results of Proxy Routing Data on MBPP (RMS distance; lower is better)*
> ||Perplexity|OOD Probe|
> |--|--|--|
> |Mistral-7B|0.0301|0.0045|
> |Qwen3-0.6B|0.0095|0.0330|
>
>
> **[Q-2] No Evaluation of Multi-LLM Routing, such as RouterBench: Cost and Generalization to New Domains**
>
> **[A-2]** Unlike the settings in RouterBench, which focus on multi-LLM routing using an external router to select the most appropriate LLM on large-scale servers for a given query, our work addresses a different stage of the routing pipeline: deciding whether to process a query locally on the edge device with an SLM or to offload it to a more powerful LLM in the cloud servers. Multi-LLM routing is outside the scope of our current on-device routing scenario, but it represents a valuable follow-up step once the offloading decision has been made. In other words, after a query is determined to route to the cloud from edge device, multi-LLM routing strategies, such as the baselines in RouterBench, can be applied to further optimize inference efficiency and budget.

---

> > ### Comment · Reviewer_2XVr · 2025-08-03
> >
> > Thanks for the clarification. I will keep my positive score.

---

### Official Review · Reviewer_nYSj · 2025-07-02

**Clarity:** 4
**Significance:** 2
**Originality:** 2
**Rating:** 4
**Confidence:** 2

**Summary:**

The paper evaluates 8 uncertainty quantification (UQ) methods across 14 datasets to assess how well uncertainty aligns with prediction correctness in SLMs, which is crucial for effective routing. Based on the finding from the benchmarking, the authors introduced a proxy routing data construction pipeline and a hold-out dataset to generalize routing strategies without prior knowledge of new downstream data.

**Questions:**

Already stated in weakness.

**Ethical Concerns:**

["NO or VERY MINOR ethics concerns only"]

**Final Justification:**

I am satisfied with the reviewer's responses. They have addressed my concerns regarding generalizability. However, the particular work itself  may have lack of novelty or innovation. I am updating my score accordingly.

**Limitations:**

No.

**Quality:**

2

**Strengths And Weaknesses:**

Strengths

The paper provides a rigorous benchmark involving 8 UQ methods, 8 SLMs, 2 LLMs, and 14 datasets, reflecting real-world deployment diversity and offering strong empirical support for their claims. The work addresses a real and growing challenge — balancing accuracy and cost in edge AI deployments—and provides actionable insights that help minimize LLM calls while preserving performance. Finally, the introduction of a proxy routing dataset to predict routing behavior in unseen tasks is both novel and practical, especially for scenarios where data collection is costly or infeasible.

Weaknesses

Authors could provide in-depth analysis of why certain UQ methods fail on specific datasets or tasks, which would be valuable for future improvements.
Despite the generalization claims, the proxy routing approach is still indirectly tuned using benchmarked datasets, which may bias the proxy data toward known distributions.
The pipeline for generating proxy routing data is not deeply explained in terms of how it adapts to drastically different downstream domains (e.g., multimodal tasks, non-English text), which limits its universality.
Can the authors propose some guidelines from the observations?

---

> ### Author Rebuttal · Authors · 2025-07-31
>
> Thank you for your tremendous efforts in reviewing our paper. We would like to further answer your questions as follows.
>
> **[Q-1] Need More Analysis of Why Certain UQ Methods Fail on Certain Datasets/Tasks**
>
> **[A-1]** The effectiveness of uncertainty-based routing relies heavily on the alignment between uncertainty scores and prediction correctness, which is a point we emphasize and validate in Section 3 (especially shown in Figure 1 for alignment assessment). A well-calibrated uncertainty score does not guarantee strong routing performance, regardless of how powerful the underlying LLMs are or how simple the tasks may seem. Misalignment between uncertainty scores and actual prediction correctness is one of the key factors leading to routing failures across different datasets/tasks. Furthermore, the distribution of correct predictions varies significantly depending on the properties of the LLMs, such as pretraining data, decoding strategies, etc. In addition, different uncertainty estimation methods can yield varying levels of alignment, further influencing routing performance across different LLMs and datasets.
>
>
> **[Q-2] Despite the generalization claims, the proxy routing approach is still indirectly tuned using benchmarked datasets, which may bias the proxy data toward known distributions**
>
> **[A-2]** We would like to respond from two perspectives:
> - First, our proxy routing approach is not parameterized, which is an untrainable method by natural. The proxy datasets are used for routing curve prediction.
> - For known distribution settings, we have considered the "partially in-domain" settings to assess the performance of proxy dataset. The domain of the proxy dataset in this settings may partially overlap to targeted testing data. The results are shown in Figure 5 and Appendix, indicating good performance in terms of using proxy data for routing curve prediction.
>
>
> **[Q-3] The pipeline for generating proxy routing data is not deeply explained in terms of how it adapts to drastically different downstream domains (e.g., multimodal tasks, non-English text), which limits its universality. Can the authors propose some guidelines from the observations**
>
> **[A-3]** In the main text of our paper (Figure 6), we evaluate Proxy Routing Data in a **highly OOD** setting: we construct proxy data only from commonsense‑reasoning and conversational/contextual datasets and evaluate on math (AQuA), routing from four SLMs to DeepSeek‑R1. In this case, our method still achieves competitive performance, demonstrating superior generalization.
>
> To further address the concern, we also evaluate on **translation** using **WMT24++ (English → Mexican Spanish)**, where Spanish does not appear in any other datasets used to build the proxy. We compute the average root mean squared (RMS) distance between the routing curve derived from the full downstream dataset and the curve obtained using our method. Lower RMS distances indicate better performance. Below we show results when routing from Mistral‑7B and Qwen3‑0.6B to GPT‑4.1 mini, using two UQ methods (OOD Probe and Perplexity). We observe that **our method predicts accurate routing thresholds without accessing the dataset at all**, obtaining small RMS distances.
>
> *Table 2: Results of Proxy Routing Data on WMT24++ (RMS distance; lower is better)*
> ||Perplexity|OOD Probe|
> |--|--|--|
> |Mistral-7B|0.0269|0.0175|
> |Qwen3-0.6B|0.0075|0.0141|

---

> ### Comment · Area_Chair_VVoa · 2025-08-05
>
> The author-reviewer discussion is coming to a close. Would you please go through the authors' rebuttal ASAP, highlight any remaining concerns (if any) so that the authors have sufficient time to respond to them, and adjust your review/rating accordingly?
>
> Thanks!
>
> Your AC

---

### Official Review · Reviewer_Qyok · 2025-07-05

**Clarity:** 3
**Significance:** 3
**Originality:** 2
**Rating:** 3
**Confidence:** 3

**Summary:**

The authors consider a setting in which small language models (SLMs) attempt to answer a question first, and then defer to a large language model (LLM) if the estimated confidence in their answer is too low.

A routing curve describes the performance of such a system for a given SLM-LLM pair as one uses a specific uncertainty quantification (UQ) metric to select which examples to escalate to the LLM. The curve shows how tweaking the proportion of examples in a dataset passed to the LLM (from none to some to all) changes the performance of the system on the entire dataset.

The authors first evaluate several UQ metrics to see which performs better at selective prediction on different datasets (measurable by AUC ROC). Then, they produce routing curves to see which UQ metrics perform best at routing for each SLM & dataset.

Finally, the authors argue and present evidence that the confidence distribution of UQ over a dataset typically depends on the UQ method and the SLM, but **not the dataset** -- arguing that the routing curve for a new dataset can be predicted by the routing curve on data from other datasets on the same SLM&UQ. They formalize this notion with Algorithm 1, a method of estimating routing curves with "proxy routing data" from other datasets.

**Questions:**

Most of my questions are addressed in the Weaknesses section above. Mostly, I am concerned about the necessity and utility of the proxy routing data concept--as well as with the novelty of this work and the resolution of this work's differing conclusion with respect to OOD routing as compared to other cited works.

**Ethical Concerns:**

["NO or VERY MINOR ethics concerns only"]

**Final Justification:**

At this time, I still have concerns behind novelty of the work--and motivation of the binning procedure which represents a large portion of the work's contribution.

I maintain my score.

**Limitations:**

The authors quickly address potential negative social impact of their work. I am inclined to agree that there is not much such impact.

However, I think the authors might address in more detail address a few more limitations of their analysis, such as:

* The cost of producing the routing dataset, including assigning buckets, is at least one inference forward pass + UQ forward pass in the full routing set before sampling 10%
* The routing analysis does not consider tweaking the LLM as much as it considers tweaking the SLM. This is another possible axis of analysis.

**Paper Formatting Concerns:**

Anonymous code link in abstract links to an anonymized repo that seems to link directly to an arxiv link for this paper. I did not follow the arxiv link so as to not compromise my review, but this seems to violate the policy:

```
All submissions must be anonymized and may not contain any identifying information that may violate the double-blind reviewing policy. This policy applies to any supplementary or linked material as well, including code. If you are including links to any external material, it is your responsibility to guarantee anonymous browsing.
```

**Quality:**

2

**Strengths And Weaknesses:**

Strengths:

**S1**

Experiments are sufficient in scale and execution for this type of paper. They are also presented in a sufficient way to argue the main points. The breadth of UQ methods examined is sufficient, and several different SLMs are explored. Contribution as a benchmark is strong.

**S2**

Writing is generally clear. I do have an issue with the section 4; I will elaborate later.

Weaknesses:

**W1**

This field is not new; there are many works (acknowledged in this paper draft) that have used uncertainty for SLM-->LLM routing--and this paper's main methodology is mostly non-novel.

One of the main ways in which this work is justified as distinct compared to these prior works is the presence of improved guidelines for better dataset/domain generalization. However, it seems that the main conclusion of this work with respect to generalization is that changing the dataset actually does NOT present much of an obstacle for generalization--and I find the proxy routing data contribution slightly overcomplicated (see my W3).

**W2**

The paper argues that other works "face significant challenges when encountering new downstream tasks, as such data falls outside the distribution of the existing training data" (133-135?). I find it not fully explained or explored why this paper's experiments, which again use a very standard setup, did not result in the same cross-domain issues that this paper argues these other papers displayed. It is certainly possible that specific tweaks in the setup led to these differences--but I think understanding and explaining any such differences is a key part of the remit of this work.

**W3**

The proxy routing contribution is presented in 4.1 and Algorithm 1. I do not find the notation in Algorithm 1 very well-explained (what is $\mathbf{X}_i$? What is $j$?) and furthermore, I think the presentation of a proxy routing dataset to be a slightly overcomplicated stratified random sample with no real justification. Why not just sample 10% of the proxy samples instead of binning and sampling? Is there a meaningful difference in the results?

If there is such a difference, would we not just counteract it by sampling more than 10%? If the justification on subsampling 10% is to save on the cost of inference when producing routing curves, don't we still have to do forward pass inference and UQ inference on every example to put it in the correct bin before sampling -- meaning that we still have done inference on every example?

**W4**

Sections 2 and 5 occupy a great deal of the work. Section 2 mostly deals with in-depth literature review (explaining each type of UQ), and Section 5 deals mostly with speculation about future research directions. Either or both could be sacrificed, condensed, or relegated to the appendices in favor of more in-depth explanation of the novelty of this work and differences between this work and other work's findings.

---

> ### Author Rebuttal · Authors · 2025-07-31
>
> Thank you for your tremendous efforts in reviewing our paper. We would like to further answer your questions as follows.
>
> **[Q-1] Lack of Discussion of Distinct Contributions of Our Work**
>
> **[A-1]** We would like to clarify that there are two non-trivial contributions in our paper stated as follows:
> - First, benchmarking routing performance based on uncertainty scores. This is a non-trivial result (we are glad that the Reviewer acknowledged our contribution in S1), as well-calibrated uncertainty scores do not always theoretically correlate with correctness [1, 2]. Therefore, a comprehensive benchmark and analysis are necessary, as previous work has often overlooked the importance of establishing fair and valid result comparisons.
> - Second, we propose a proxy dataset construction method to predict the routing curve without the need for input queries, which enables efficient analysis of routing performance–cost trade-offs. In contrast, existing work [3, 4] typically relies on the **(1) full set of test queries to construct the routing curve** and **(2) they observe these trade-offs between large-scale LLMs**. Our work generates a proxy dataset and tests its ability for assisting SLM routing without any user queries. The constructed data can accurately predict the routing curve, allowing users to anticipate which UQ method and threshold offer the best trade-off even on SLM settings.
>
> [1] Huang, et al. "Look before you leap: ...." arXiv 2023.
>
> [2] Yona, et al. "Can Large Language Models Faithfully ...?." arXiv 2024.
>
> [3] Ong, et al. "RouteLLM: Learning to Route LLMs from Preference Data." ICLR. 2024.
>
> [4] Hu, et al. "Routerbench: A benchmark for multi-llm routing system."  arXiv 2024.
>
>
> **[Q-2] Why Prior Work Faces Challenges on New Downstream Datasets**
>
> **[A-2]** We would like to clarify from two points:
> - Classifier-style routers trained on historical prompt–performance data often struggle on new downstream tasks because the input distribution and the relative strengths of candidate LLMs shift. RouterBench [1] explicitly runs "out‑domain" experiments to test cross‑task transfer, observing that some routers fail to generalize to new tasks or new models.
> In contrast, our benchmark focuses on routing methods based on uncertainty quantification (UQ), which do not rely on training on historical prompt–performance data. However, even for training‑free UQ methods (e.g., average token probability, verbalization), they also face a challenge: how to set routing thresholds for unseen datasets to acheive effecitve routing performance. Our proposed Proxy Routing Data approach addresses this by effectively estimating routing curve for new downstream data.
> - In our experiments, we test the Proxy Routing Data under the setting of "fully out-of-domain" to showcase the generalization ability on the challenges faced by the prior work. The results show that Proxy Routing Data can help predict the routing curve accurately without needing any out-of-domain or new downstream data.
>
> [1] Hu, et al. "Routerbench: A benchmark for multi-llm routing system." arXiv 2024.
>
>
> **[Q-3] Unclear Notations in Algorithm 1**
>
> **[A-3]** We are sorry for the confusion. $X_i$ denotes the subset of proxy data $X$ followed the confidence score distribution $\\{F_{\mathbb{D}}\\}_{i=1}^{M}$ of $i$ datasest $D_i$. The element $x_j \in X_i$ denotes the $j$-th element in $X_i$. Thus, we can say, for every element $x_j \in X_i \subset X$. We will include this in our next version of paper.
>
> **[Q-4] Justification of Sampling in Proxy Routing Data Construction**
>
> **[A-4]** Our goal focuses on the routing configuration of on-device SLM. Unlike prior work conducted on large-scale server environments, on-device settings typically involve limited storage and computational resources, making it infeasible to store large-scale datasets on edge devices. Therefore, we evaluate routing curve prediction using a small proxy dataset designed to simulate real-world on-device scenarios. While it is possible to use 100% of the data for routing prediction under unconstrained resource conditions, such an approach contradicts the resource-limited assumptions of on-device SLMs explored in this work, which is the reason that our paper leverage 10\% of data as an example. To better approximate the distribution of the original data, we adopt binned sampling, which ensures that the sampled proxy data more closely reflects the characteristics of the full population.
>
> To further address the reviewer’s concern, we varied the sampling ratio used to construct proxy routing data while fixing the number of bins at 30. On HellaSwag, we routed from three SLMs to GPT‑4o‑mini using the OOD Probe. We report the average RMS distance between the Routing Ratio vs. Accuracy curves obtained with thresholds from the full downstream dataset and those derived from our method (lower is better). Results (in Table 1) show that our proxy data construction pipeline maintains **its ability to predict routing curves even with a very small sample size** (only sampling **1%**), supporting its suitability for on‑device scenarios.
>
> *Table 1: Varying Sampling Ratio (Fixed #Bins = 30)*
>
> |Ratio|Llama-3.2-3B-Instruct|Mistral-7B-Instruct-v0.3|Llama-3.1-8B-Instruct|
> |--|--|--|--|
> |0.01|0.025|0.020|0.023|
> |0.05|0.025|0.021|0.023|
> |0.1|0.025|0.020|0.023|
>
>
> **[Q-5] Sections 2 and 5 Can Be More Condensed**
>
> **[A-5]** We will further condense Sections 2 and 5 (~ 1.5 pages now) and include the discussions we had during the rebuttal in our next version. In Section 2, we include the introduction of method in our benchmarking settings, while in Section 5, we include the discussion of using proxy data to on-device scenarios, which are essential to highlight our contributions.
>
> **[Q-6] Sufficient Tweaking of SLMs; Insufficient Tweaking of LLMs**
>
> **[A-6]** We focus on deploying SLMs on edge devices, which motivates our emphasis on routing from SLMs to LLMs. Routing between LLMs falls outside the scope of this paper. We appreciate the reviewer’s insightful comment and will consider this as a promising direction for future work.
>
> **[Q-7] The cost of producing the routing dataset is at least one inference forward pass + UQ forward pass in the full routing set**
>
> **[A-7]** All routing data can be precomputed before deploying for edge device routing curve prediction, meaning that the cost of generating the routing dataset, consisting of one or two forward pass for inference, is incurred only once offline, and does not affect the efficiency during on-device inference serving.

---

> > ### Comment · Reviewer_Qyok · 2025-08-06
> >
> > [A-1] Noted, thank you. I appreciate the clarification.
> >
> > [A-2] Thank you. It seems the main distinction is that classifier-style routers suffer OOD, but UQ-based routers do not. This section 2.2 might be well-served by addressing other work on UQ-based routers. Perhaps it would be worth addressing any previous findings regarding generalization performance in previous works leveraging UQ-based routers (or specifically noting the absence of such findings).
> >
> > [A-3] Thank you.
> >
> > [A-4] [A-7] I am struggling to reconcile these two answers. The motivation of limited on-device storage is valid. I accept that "all routing data can be precomputed" for a given SLM/UQ method pair "before deploying for edge device routing curve prediction"--but maybe do not quite grasp why it is then necessary to cache the data examples on the edge device at all instead of just caching the routing curve itself. It seems that [A-4] assumes that the forward pass of the routing proxy data happens on-device while [A-7] assumes it happens once globally.
> >
> > I acknowledge it may seem that I am nitpicking for no reason. The reason that this matters is that the binning procedure is ONLY relevant if somehow we CAN do the O(full dataset) forward pass to assign bins only once--but somehow MUST repeat the O(subsampled proxy data) forward pass inference cost on every edge device. In any other case, we would be better off just computing the full routing curve on all of the data.
> >
> > That is to say, I still find the notion of a "proxy routing data construction pipeline" to be overcomplicated--and a bit overstated.
> >
> > [A-5] Thank you.
> >
> > [A-6] My apologies, I perhaps was not clear enough. My issue is not that routing from LLM --> LLM was not explored. Instead, I was curious about the similarity of routing curves/utility of proxy data when routing from different SLMs to the same LLM--instead of from the same SLM to different LLMs. I do acknowledge, upon some further reflection, that the characteristics and behavior of the UQ method are much more closely tied to the SLM, so I frankly am not sure that this direction of exploration would yield obviously low-hanging fruit. I suppose it would be nice to confirm, but it's not a glaring miss for the paper.
> >
> > Things are looking good. I appreciate the clarification of the contribution. I would thoroughly recommend clarifying the wording in 2.2 slightly. Right now it reads a bit like this paper is the first to propose that UQ is useful for routing, as opposed to other papers which only use routing classifiers.

---

> > > ### Author Response · Authors · 2025-08-07
> > > **Response to the Official Comment by Reviewer Qyok. Thank you for your time and comments!**
> > >
> > > Thank you for your detailed response; we really appreciate the tremendous effort you put into reviewing our paper. Below, we address your remaining concerns.
> > >
> > > **[FQ-1] Follow-up questions about [A-2]**
> > >
> > > **[FA-1]** Thank you for your valuable suggestions on improving Section 2.2. We further emphasize that **Section 2.1** provides a taxonomic overview of UQ methods for language models, and these methods are natural routers.
> > >
> > > All UQ-based routers suffer from a critical generalization issue: they trigger routing when a query’s uncertainty score exceeds a chosen threshold, but that **threshold is unknown on unseen datasets**. We cannot know, a priori, what threshold to use for safe, out-of-the-box routing on new data. In our work, we introduce Proxy Routing Data, which addresses this problem. We would like to mention that this generalization issue is highlighted in the Abstract, Introduction, and Section 4. We will also make these limitations of prior UQ-based routing methods more explicit in the revised Section 2.2.
> > >
> > > **[FQ-2] Follow-up questions about [A-4] & [A-7]**
> > >
> > > **[FA-2]** Thank you for the follow-up question and comments. Caching the raw, binned proxy examples on each edge device remains advantageous whenever the uncertainty-quantification (UQ) metric must be updated rapidly to reflect user inputs or distributional shifts/domain drifts. We can reconstruct the new routing curve based on the stored proxy data and quickly adapt to new distribution without sending user data back to protect privacy. Because the proxy set was stratified bin-by-bin from the full corpus, it provides a statistically sound miniature of the production data.
> > >
> > > - Caching a 0.5 MB, ~2,000 example proxy set on each device gives us a flexible, statistically grounded way to recalibrate routing curves under both routine metric swaps and rarer distribution shifts. For everyday updates (e.g., switching from log-probability to entropy or ensemble variance) re-scoring and re-bucketing the cached examples preserves ≈ 65 samples per bin (under the 30-bin setting), keeping the per-bin standard error ≤ 0.062 and the overall RMSE per example deviation from the full-corpus curve ≤ 1.12 % with zero download cost.
> > >
> > > Here we provide the derivation of the ≤ 0.062 per-bin SE and RMSE ≤ 1.12%, preserving ≈ 65 samples per bin under 30-bin setting.
> > >
> > > > For each bin, the accuracy indicator for UQ-based routing setting $Z_i=\mathbf 1\\{y_i=\hat y_i\\}$ is Bernoulli with success probability $p$.
> > > >
> > > >    $$
> > > >      \operatorname{Var}\bigl[\hat a_b\bigr]=\frac{p(1-p)}{k}\le\frac{0.25}{k}\quad(\text{max at }p=0.5).
> > > >    $$
> > > >
> > > > Then, we calculate the *Per-bin SE*
> > > >
> > > >    With $k\approx 65$ examples per bin,
> > > >
> > > >    $$
> > > >      \text{SE}_b\le\frac{0.5}{\sqrt{k}}=\frac{0.5}{\sqrt{65}}\approx 0.062.
> > > >    $$
> > > >
> > > > Lastly, *RMSE* can be calculated as: when there are $B=30$ equal-weight bins and $N=\sum_b k\approx 2\,000$ total samples.
> > > >
> > > > $$\text{RMSE}= \sqrt{\frac{1}{N}\sum_b k\,\operatorname{Var}[\hat a_b]} = \frac{0.5}{\sqrt{N}}\approx0.0112 (\text{≈ 1.12 \\%}).$$
> > >
> > > - If a major domain change empties some bins, the device detects the imbalance, then either tops up those strata with a few dozen labeled examples or, in the extreme, replaces the entire proxy file; in both cases the statistical guarantees are promptly restored, while bandwidth and storage remain a challenge for edge device routing settings.
> > > - A one-shot recalibration over the ≈ 2 000-example proxy set finishes in ≈ 0.6 s and consumes ≈ 0.4 J on the Pixel 7’s Tensor G2 NPU (≈ 200 µJ per inference [1]). An order of magnitude less than the 5–15 J typically required to download a 0.5 MB patch over LTE in realistic network conditions [2]. Regardless, resending the user data back to the server may result in privacy issues. Thus, storing the proxy data would be a good option.
> > >
> > > We will update the discussion and statistical derivation into the next version of our paper. Thanks!
> > >
> > > [1] Akin, et al. "Searching for efficient neural architectures for on-device ML on edge TPUs." Proceedings of the IEEE/CVF Conference on Computer Vision and Pattern Recognition. 2022.
> > >
> > > [2] Caiazza, et al. "Energy consumption of smartphones and IoT devices when using different versions of the HTTP protocol." Pervasive and Mobile Computing 97 (2024).
> > >
> > > **[FQ-3] Follow-up questions about [A-6]**
> > >
> > > **[FA-3]** Thank you for the clarification. The behavior of UQ is more tightly coupled with the SLM, which is why our primary analysis emphasizes routing from the same SLM to one/different LLMs. Our rationale is practical: in on-device settings, **one typically hosts a single local SLM**, and decides whether to route a query to cloud LLMs based on latency/cost vs. accuracy. That said, our experiments **do include** the complementary direction—different SLMs routing to the same LLM. **We evaluate 12 SLMs**; by fixing the UQ method, dataset, and target LLM, we can directly compare the behaviors of different SLMs.

---

> ### Comment · Area_Chair_VVoa · 2025-08-05
>
> The author-reviewer discussion is coming to a close. Would you please go through the authors' rebuttal ASAP, highlight any remaining concerns (if any) so that the authors have sufficient time to respond to them, and adjust your review/rating accordingly?
>
> Thanks!
>
> Your AC

---

### Official Review · Reviewer_c8Rx · 2025-07-05

**Clarity:** 3
**Significance:** 3
**Originality:** 3
**Rating:** 4
**Confidence:** 4

**Summary:**

To address the problem of limited capabilities in small models and the high cost of deploying large models, this paper analyzes various existing methods for switching to large models based on the uncertainty of small models.

Specifically, the paper provides extensive analysis and experiments involving 16 models (12 SLMs, 4 strong LMs), 8 uncertainty quantification methods, and 15 benchmarks. Furthermore, the authors introduce a pipeline to construct the proxy routing data. This data can generalize its routing standards across various new downstream datasets.

**Questions:**

1. I am curious about constrained tasks like translation (e.g., WMT'14), which should arguably have lower uncertainty than open-ended tasks. How well does your main experiment and pipeline for constructing proxy routing data work on this task?



2. Your experiments find that perplexity (ppl) is a good metric. Recently, entropy has also become a very popular measure of uncertainty, even being used in reinforcement learning. Could you please supplement your results with an evaluation using entropy as an uncertainty measure?



3. The OOD Probe method is crucial for generalizing to new datasets. However, in your experiments, it was trained on significantly more data (the entirety of other datasets) than the Trained Probe (a subsample of one dataset). To conduct a fairer comparison and better quantify the trade offs, could you run an experiment where the OOD Probe is trained on a data volume comparable to that of the Trained Probe? This would help clarify the extent to which its effectiveness stems from data diversity versus data volume itself.



4. How sensitive is the final routing curve prediction to the hyperparameters of your pipeline, such as the number of bins for the distribution (M=30) and the sampling rate (10%)?

**Ethical Concerns:**

["NO or VERY MINOR ethics concerns only"]

**Final Justification:**

Thank you for your detailed rebuttal. It has alleviated most of my concerns. However, the work remains lacks a theoretical foundation. Given this remaining concern, I will maintain my score of 4 (Borderline accept).

**Limitations:**

Yes

**Quality:**

3

**Strengths And Weaknesses:**

**Strengths:**



- The main strength of this paper lies in its substantial empirical contributions. The large scale benchmark (with 16 models including 12 SLMs and 4 strong LMs, 8 uncertainty methods, and 15 benchmarks) is an important and valuable resource for the NLP community, providing a good foundation for the paper's claims and future research in this area.



- This paper is not just a simple analysis paper. In order to solve the problem of how to transfer routing standards to new and unseen tasks, the paper further proposes a pipeline to construct the proxy routing data, which can generalize its routing standards across various new downstream datasets.



- The paper is written with clarity and the open source code makes the work easier to reproduce





**Weaknesses:**



- The work suffers from a lack of deeper theoretical explanation for the empirical findings. For example, uncertainty distributions depend more on both the specific SLM and the chosen UQ method, rather than downstream data.  A more in-depth analysis will further enhance the contribution of this paper



- The training data sizes for the "trained probe" (a subsample of one dataset) and the "OOD probe" (the entirety of other datasets) are vastly different. It is unclear if the strong performance of the "OOD probe" could be maintained under a more controlled comparison with an equal amount of training data, making it difficult to assess the true cost and benefit of this approach.

---

> ### Author Rebuttal · Authors · 2025-07-31
>
> Thank you for your tremendous efforts in reviewing our paper. We would like to further answer your questions as follows.
>
>
> **[Q-1] Insufficient Theoretical Explanation of Empirical Findings**
>
> **[A-1]**  We would like to clarify that:
> - The effectiveness of uncertainty-based routing depends on the alignment between uncertainty scores and prediction correctness from SLM; however, this alignment **cannot be theoretically guaranteed by nature** [1, 2]. Thus, we provide a comprehensive empirical study to discover the pattern in our two main directions with our explanations across Section 3, 4, and 5.
> - We provide two toy examples to explain this misalignment intuitively: Assume we have a binary classification task and obtain results from two binary classification models with different predicted probabilities (see below). Example 1 achieves lower accuracy but has a lower ECE score, whereas Example 2 achieves higher accuracy but a higher ECE score. This example demonstrates that calibration metrics are not generally correlated with the prediction's correctness.
>
> ```
> Example 1:
> Bin = 2
> predicted prob. of "1" = [0.5, 0.5, 0.5, 0.5]
> ground truth = [0, 0, 1, 1]
> -> ECE(↓)= 0%
> -> accuracy (↑)= 50%
>
> Example 2:
> Bin = 2
> predicted prob. of "1" = [0.4, 0.6, 0.9, 0.9]
> ground truth = [0, 0, 1, 1]
> -> ECE(↓) = 20%
> -> accuracy (↑) = 75%
> ```
>
> [1] Huang, et al. "Look before you leap: An exploratory study of uncertainty measurement for large language models." arXiv, 2023.
>
> [2] Detommaso, et al. "Multicalibration for confidence scoring in llms." arXiv 2024.
>
>
>
> **[Q-2] Imbalanced Training Sizes Between Trained Probe and OOD Probe**
>
> **[A-2]** Thank you for raising this concern. Actually, in all experiments relevant to OOD Probe, we **explicitly controlled for training volume**: we **down‑sampled** the pool of remaining datasets so that the OOD Probe was trained on a **similar number of examples as the Trained Probe**. Specifically, rather than using all available examples from other datasets, we randomly sampled a small, fixed percentage from each to match the Trained Probe’s data volume. Under this controlled setting, it's safe to infer that the performance comes from data diversity instead of volume. We will make this important setup more explicit in the final version to avoid confusion.
>
>
> **[Q-3] Results on Translation Tasks**
>
> **[A-3]** We ran additional experiments on **WMT24++** (English to Mexican Spanish translation). Below we report ROC AUC (uncertainty–correctness alignment) for two SLMs across 8 UQ methods. Perplexity and Average Token Probability perform best for Mistral‑7B, while Jaccard Degree and Trained Probe perform best for Qwen3‑0.6B.
>
> *Table 1: Uncertainty–Correctness Alignment on WMT24++*
> ||AverageTokenProb|P(True)|Perplexity|Jaccard Degree|Verbalization-1s|Verbalization-2s|Trained Probe|OOD Probe|
> |--|--|--|--|--|--|--|--|--|
> |Mistral-7B|**0.6516**|0.5232|**0.6647**|0.5|0.5231|0.5|0.4974|0.5029|
> |Qwen3-0.6B|0.6183|0.5|0.6295|**0.7446**|0.5320|0.4507|**0.6938**|0.6130|
>
> Next, we evaluate our **Proxy Routing Data** on WMT24++. Since we are not able to provide figures like Figure 5 and Figure 6 here, we compute the average root mean squared (RMS) distance between the routing curve derived from the full downstream dataset and the curve obtained using our method. Lower RMS distances indicate better performance. Below we show results when routing from Mistral‑7B and Qwen3‑0.6B to GPT‑4.1 mini, using two UQ methods (OOD Probe and Perplexity). We observe that **our method predicts accurate routing thresholds on WMT24++ without accessing it at all**, leading to small RMS distances.
>
> *Table 2: Results of Proxy Routing Data on WMT24++ (RMS distance; lower is better)*
> ||Perplexity|OOD Probe|
> |--|--|--|
> |Mistral-7B|0.0269|0.0175|
> |Qwen3-0.6B|0.0075|0.0141|
>
>
> **[Q-4] Entropy as an Uncertainty Quantification Method**
>
> **[A-4]** Thank you for the suggestion. We have implemented **Entropy** as a new UQ method. Below we report ROC AUC (uncertainty–correctness alignment) on OpenBookQA across multiple SLMs using uncertainty derived by Entropy. Compared with Figure 1 in our paper, Entropy **performs similarly** to token/sequence‑probability methods (**Average Token Probability** and **Perplexity**): all three work well on the Llama series and are less effective on the Qwen series, obtaining similar numbers across most SLMs. We will include these results in the final version.
>
> *Table 1: Uncertainty–Correctness Alignment (ROC AUC) Using Entropy on OpenBookQA*
> ||Llama-3.2-1B|Llama-3.2-3B|Qwen2.5-7B|Mistral-7B|Llama-3.1-8B|Granite-3.1-8B|Qwen3-0.6B|Qwen3-1.7B|Qwen3-4B|Phi-4-mini-reasoning|
> |--|--|--|--|--|--|--|--|--|--|--|
> |ROC AUC|0.6666|0.7498|0.5287|0.6721|0.7271|0.5874|0.4996|0.5099|0.5638|0.6175|
>
>
> **[Q-5] Sensitivity Analysis of Hyperparameters**
>
> **[A-5]** We conducted a sensitivity analysis on both the sampling ratio and the number of bins to assess our method’s generalizability and robustness. The first table below fixes the number of bins at 30 while varying the sampling ratio; the second fixes the sampling ratio at 0.1 while varying the number of bins. These experiments were performed on HellaSwag, routing from three SLMs to GPT-4o-mini using the OOD Probe. We computed the average RMS distance between the Routing Ratio vs. Accuracy curves obtained using thresholds from the full downstream dataset and those derived from our method—lower RMS distances indicate better performance.
>
> *Table 1: Varying Sampling Ratio (Fixed #Bins = 30)*
>
> |Ratio|Llama-3.2-3B-Instruct|Mistral-7B-Instruct-v0.3|Llama-3.1-8B-Instruct|
> |--|--|--|--|
> |0.01|0.025|0.020|0.023|
> |0.05|0.025|0.021|0.023|
> |0.1|0.025|0.020|0.023|
> |0.2|0.025|0.020|0.024|
> |0.3|0.025|0.020|0.023|
>
> *Table 2: Varying Number of Bins (Fixed Sampling Ratio = 0.1)*
> |#Bin|Llama-3.2-3B-Instruct|Mistral-7B-Instruct-v0.3|Llama-3.1-8B-Instruct|
> |--|--|--|--|
> |10|0.025|0.021|0.022|
> |15|0.025|0.020|0.024|
> |20|0.025|0.021|0.024|
> |30|0.025|0.020|0.024|
> |40|0.025|0.021|0.023|
> |50|0.025|0.021|0.023|
>
> From the tables above, we can see that **our method is extremely stable** and robust against various selections of sampling ratios and number of bins. We will ensure that this robustness characteristic is highlighted in the final version of our paper.

---

> > ### Comment · Reviewer_c8Rx · 2025-08-06
> >
> > Thank you for your detailed rebuttal. It has alleviated most of my concerns. However, the work remains lacks a theoretical foundation. Given this remaining concern, I will maintain my score of 4 (Borderline accept).

---

> ### Comment · Area_Chair_VVoa · 2025-08-05
>
> The author-reviewer discussion is coming to a close. Would you please go through the authors' rebuttal ASAP, highlight any remaining concerns (if any) so that the authors have sufficient time to respond to them, and adjust your review/rating accordingly?
>
> Thanks!
>
> Your AC

---

### Note · Authors · 2025-08-16

We sincerely thank the AC and reviewers for their time and constructive feedback. We are pleased to find that they recognize our substantial empirical contributions (c8Rx, Qyok, nYSj, 2XVr), the novelty and practicality of our proxy data construction pipeline (c8Rx, nYSj), the clarity of our writing (c8Rx, Qyok, 2XVr), and the breadth and rigor of our benchmark across models, datasets, and UQ methods (c8Rx, Qyok, nYSj, 2XVr).

Following the rebuttal, we are encouraged that two reviewers (2XVr and c8Rx) maintained their positive assessments. While we were also looking forward to further response from Reviewer nYSj, no additional feedback was received.

We believe our rebuttal has effectively clarified and resolved most points. Below, we summarize the key aspects of our responses.

- **Lack of theory supports.** We would like to emphasize that the effectiveness of uncertainty-based routing between SLMs and LLMs fundamentally depends on the alignment between uncertainty scores and prediction correctness from SLMs. However, such alignment cannot be theoretically guaranteed [1, 2], which underscores the necessity of profiling and benchmarking. To address this, we present a comprehensive empirical study that systematically explores the patterns in our two main directions, with detailed explanations provided in Sections 3, 4, and 5. Therefore, the absence of a formal theoretical guarantee should not be considered a weakness of our paper. Moreover, during rebuttal, we complement our empirical findings with a theoretical analysis that supports both the effectiveness and necessity of proxy data generation. We will include this analysis in the next version of our paper.

- **The motivation of proxy data construction.** We have provided both discussion and theoretical analysis to support the necessity and effectiveness of using proxy data for predicting routing curves on edge devices. In addition, we highlight the advantages of storing proxy data on edge devices, as it enables fast and statistically reliable recalibration under both routine metric updates and rare distribution shifts, maintains low per-bin error with minimal storage, avoids the high energy and bandwidth costs of downloading patches for edge devices, and mitigates potential privacy risks from resending user data back to the server.

[1] Huang, et al. "Look before you leap: ..." arXiv, 2023.

[2] Detommaso, et al. "Multicalibration ... LLMs." arXiv 2024.

---

### Decision · Program_Chairs · 2025-09-17

**Decision:**

Reject

**Comment:**

A summary of the strengths and weaknesses based on the reviews and rebuttal (including the follow-up discussion among the reviewers and authors) is provided below.


**STRENGTHS**

(1) The main contribution of this work lies in providing a comprehensive benchmark and extensive empirical findings with many SLMs and LLMs, UQ methods, and datasets (Reviewers c8Rx, nYSj, and Qyok).

(2) The authors have proposed a new pipeline to construct the proxy routing data to generalize its routing strategy without prior knowledge of new downstream data (Reviewer c8Rx).

**WEAKNESSES**

Several major technical concerns remain, requiring the authors to address them:

(1) A number of reviewers (Reviewers Qyok and nYSj) have voiced that novelty and positioning of this work remain unclear relative to the existing works, as explained in their reviews.

A follow-up question from Reviewer Qyok in the latest response, which the authors might not be able to see, is on whether this work is claiming to be the first to try UQ for routing and the first to observe issues with generalization.

(2) This work lacks an in-depth, rigorous analysis to explain and understand the observations/findings as well as guidelines, as raised by Reviewers nYSj, Qyok, and c8Rx and discussed in their reviews. For example, considering your finding of misalignment between uncertainty scores and prediction correctness, would there be certain simplified yet practical task settings that reliably and consistently exhibit strong alignment? Why?

(3) The rationale of using binned sampling in constructing the proxy routing data and caching it remains unclear (Reviewer Qyok). In the rebuttal, the authors have brought up its advantage for distributional shifts/domain drifts, which does not seem to be described in the main paper. It is also not clear whether the authors are proposing a different method in the rebuttal to handle distributional shifts/domain drifts.

Minor note: There are unclear notations in Algorithm 1, which have been clarified in the authors' rebuttal to Reviewer Qyok.


From the above, the cons outweigh the pros. A major revision of the paper is necessary.

The authors are strongly encouraged to revise their paper based on the above feedback and that of the reviewers.